# Ultra-Dispersed Powders Produced by High-Temperature Shear-Induced Grinding of Worn-Out Tire and Products of Their Interaction with Hot Bitumen

**DOI:** 10.3390/polym14173627

**Published:** 2022-09-01

**Authors:** Vadim Nikol’skii, Tatiana Dudareva, Irina Krasotkina, Irina Gordeeva, Alexandre A. Vetcher, Alexander Botin

**Affiliations:** 1N.N. Semenov Federal Research Center of Chemical Physics, Russian Academy of Sciences, 4, Kosygin St., 119991 Moscow, Russia; 2Institute of Biochemical Technology and Nanotechnology, Peoples’ Friendship University of Russia (RUDN), 6 Miklukho-Maklaya St., 117198 Moscow, Russia; 3Complementary and Integrative Health Clinic of Dr. Shishonin, 5 Yasnogorskaya St., 117588 Moscow, Russia; 4N.V. Sklifosovsky Institute of Emergency Medicine, 3 Bol’shaya Sukharevskaya Sq., 129090 Moscow, Russia

**Keywords:** powder elastomeric modifiers, high-temperature shear-induced grinding, fractality, bitumen, rapid breakdown, swelling

## Abstract

Structural features of crumb rubber (CR) particles obtained by grinding on rollers and ultra-disperse powder elastomeric modifiers (PEM) obtained by high-temperature shear-induced grinding (HTSG) of CR or co-grinding with butadiene styrene thermoplastic elastomer (SBS) have been studied by electron and optical microscopy methods. Samples of modified bitumen were obtained at different mixing times (1–40 min) in a wide temperature range (120–180 °C). The products of interaction of PEM with hot bitumen precipitated on filters when washed with solvent from modified bitumen (MB) were studied by scanning electron microscopy (SEM). The self-similarity PEM particles and PEM breakdown fragments in bitumen up to the size of 100–200 nm were noted. The rapid (for 1 min) decomposition of PEM particles into fragments is shown, which is due to the specific structure formed as a result of HTSG. It has been suggested that this fragmentation may be caused by bitumen penetrating deep into the porous particle and breaking it, due to differently directed swelling pressure and precede the classical swelling associated with the penetration of solvent between rubber macromolecules, or occur concurrently with it.

## 1. Introduction

Rational use of secondary resources, among which worn-out tires are of particular interest, is an important task [1].

Among the directions of use of crumb rubber, obtained from worn-out tires, the most interesting is its use in pavement [2,3,4]. The use of crumb rubber as a modifier of bitumen and asphalt mixtures in order to increase their lifespan has great potential in the road industry. Combining the problems of recycling used tires and increasing the durability of road pavements opens the transition to a closed-loop economy, which allows us to reduce the consumption of primary natural resources.

Recycling of worn-out tires is mainly carried out to the size of crumb rubber (CR) of a few millimeters, as further reduction in the size requires significant energy costs. The method of high-temperature shear-induced grinding (HTSG) allows obtaining ultra-disperse powders from CR at specific energy costs not exceeding 150 watts/kg. The process of HTSG is realized in specialized equipment—rotary dispergators. During grinding, the material is subjected to a pressure of about 20–30 MPa and modulated shear stresses at temperatures close to the devulcanization temperature of rubber. At the same time, in the process of grinding, there is no release of reinforcing filler (carbon black), which is used in the production of tire rubber. The so-called active powder of discretely devulcanized rubber (APDDR) is formed. It was shown earlier that the morphology of APDDR particles distinguishes from the morphology of crumb rubber particles obtained by processing on rollers or by cryogenic milling [5].

The use in the world practice of modified bitumen containing in their CR and different polymer [6,7] has motivated the development of hybrid powders produced by high-temperature co-grinding together CR and SBS under shear-induced stress.

Many works are devoted to studying the effect of the grinding method of worn-out tires, CR size, and concentration on the properties of bitumen and asphalt concrete, of which we can mention the latest and note that interest in this topic is high [8,9,10,11,12,13,14,15].

Researchers who study the interaction of crumb of various rubbers with solvents typically note three possible processes: swelling due to solvent absorption by the vulcanizes (the most frequent phenomenon); extraction, i.e., dissolution and elution of certain rubber components into the solvent, associated with a decrease in rubber volume (less frequently); chemical interaction of the solvent with the rubber (rarely). In real conditions, these three processes may occur simultaneously [16,17].

For the stage of bitumen absorption by crumb rubber, which is determined by the composition, chemical and physical properties of bitumen and CR, as well as the temperature and duration of mixing mode [16,18], the following processes are usually considered [16,19,20,21,22,23]:-Rubber particles swelling in the maltene (aromatic) fraction of bitumen, i.e., the process of their volume expansion due to the absorption of light fractions from bitumen, meanwhile the penetration rate of the relevant bitumen fractions into the volume of the rubber particle is determined by the bitumen viscosity;-Devulcanization, i.e., breaking the C–S and S–S bonds in the vulcanized material;-Depolymerization, i.e., weakening of intermolecular bonds and breaking of C–C bonds in the rubber molecules.

The bitumen’s chemical nature determines the equilibrium swelling, and the bitumen’s viscosity determines the penetration rate of the appropriate bitumen fractions into the volume of the rubber particle [24]. As one raw material for the production of crumb rubber is unsorted worn tires, so the composition of crumb rubber is represented by a set of different rubbers, among which the most common: isoprene, butadiene-styrene, butadiene, and natural rubber. The tendency of rubber of different natures to swell in aliphatic hydrocarbons is somewhat different, which may lead to the development of additional internal stresses in the swelling layer.

If the temperature of the preparation of the rubber–bitumen mixture is high enough and the mixing time is long enough, devulcanization and depolymerization processes may lead to complete rubber disintegration, which reduces the modifier effect on bitumen properties [25,26].

The interaction kinetics of tire rubber-based modifiers with bitumen and the size of the interaction products determine the directions for using the modifier in the road industry—by the “wet” process (a separate technological stage of preparation of modified binder), or by the “dry” process, introducing the modifier directly into the mixer at the time of preparation of asphalt mixture.

Typically, the process of combining bitumen and rubber crumbs to improve the bitumen rheology is carried out by mixing at a temperature of at least 180 °C and for at least one hour. It is shown that the bulk of the introduced rubber particles, which can be traced by washing them from the bitumen, does not undergo significant changes during long-term contact with hot bitumen [27,28], unlike the APDDR, which, as shown in [5], is able to break up in hot bitumen into particles of micron and submicron size. This is important because it is shown that even a slight (5%) increase in the content of rubber particles with a size of less than 75 microns leads to a noticeable increase in the fatigue strength of modified bitumen [27].

The development of high-temperature shear-induced grinding technology and the development of a new design of rotary dispergators carried out by the authors allowed us to provide stable quality powder elastomeric modifiers on an industrial scale. This paper’s goal was to investigate the temperature–time intervals of PEM interaction with bitumen in order to work out recommendations for the use of PEM in the production of various asphalt concrete, crushed-mastic mixtures, and modified bitumen binder.

## 2. Materials and Methods

### 2.1. Materials

The following materials were investigated in this work:-Crumb rubber (CR) of unsorted worn tires with a size of less than 1 mm, obtained by grinding on rollers. The specific surface determined by the BET method at T = 77 K was about 0.3 m^2^/g.-Industrial linear butadiene styrene thermoplastic elastomer SBS L 30-01 (styrene content 30%, particle size about 1–2 mm (Sibur LLC, Moscow, Russia);-Powder elastomeric modifiers (PEM): APDDR and hybrid powder, obtained by high-temperature shear-induced grinding of CR in a rotary dispergator. APDDR was produced by grinding CR and hybrid powder by co-grinding of 80 wt.% CR and 20 wt.% SBS L 30-01. The APDDR and hybrid powder were homogeneous black powders less than 0.63 mm in size. The specific surface determined by the BET method at T = 77 K was at least 0.45 m^2^/g;-Modified binders prepared by mixing 10 ÷ 20 wt.% APDDR with 90 ÷ 80 wt.% bitumen heated to 120–180 °C with a paddle stirrer (IKA HB10 DIGITAL) for 1–40 min. Blown bitumen grade BND 60/90 with penetration of 60 dmm at 25 °C was used as a basis for the preparation of modified binders.

### 2.2. Research Methods

Research on the surface morphology of CR, SBS, APDDR, and hybrid powder particles and breakdown products of PEM in bitumen was carried out on an electron microscope “JEOL Jsm-6380 LA” (JEOL Corp., Tokyo, Japan). A “Micros 5000” optical microscope (Micros Gmbh., Wien, Austria) was also used.

### 2.3. Preparation of Samples for Electron Scanning Microscopy

#### 2.3.1. CR, APDDR, and Hybrid Powder Particles

The powder sample was thoroughly mixed and scattered as a layer on the surface of a paper sheet. A double-sided adhesive tape placed on a metal table was pressed onto the thoroughly mixed thin layer of the powder sample. To improve the microphotography clarity and to increase the charge dripping from the material under study, a 100–300 Å layer of gold was sprayed on the surface of the samples using a “Fine coat” “JFC-1100” (JEOL Corp., Tokyo, Japan).

#### 2.3.2. Breakdown Products (Fragments of Original Particles) of APDDR in Hot Bitumen

To study the products of interaction of APDDR with hot bitumen (fragments of original particles), a modified binder was washed on the microfilter with solvents (Stoddard solvent, chloroform, petroleum solvent “Nefras S2-80/120”). Cellulose filters with a pore size of about 2–3 μm and polycarbonate microfilters with an average pore size of about 100–200 nm were used.

## 3. Results and Discussion

### 3.1. Analysis of the Morphological Characteristics of CR and PEM

The crumb rubber (CR) obtained by roll (cascade) grinding with a particle size of less than 1 mm was used as the original material for high-temperature shear-induced grinding processing. Typical CR images (overview and individual particles) are shown in Figure 1.

In Figure 1A–F, SEM images of crumb rubber particles obtained by roll grinding, one can see both small particles of 5–20 μm and a significant number of large particles with sizes exceeding 500 μm. Large CR particles are characterized by sharp edges with smooth or weakly developed surfaces of facets, indicating that grinding occurred at temperatures up to 100 °C and at low shear strain. The finer particles are characterized by different ratios of smooth and developed surface areas. Areas with a developed surface of 10–15 μm are observed, but their number is relatively small, and rather smooth (flat) surfaces of various dimensions predominate in the mass.

Figure 1G–H shows the general view of APDDR particles obtained by high-temperature shear-induced grinding of CR in a rotary dispergator at 140–160 °C in the grinding zone. Analysis of microphotographs shows a significant reduction in the number of particles larger than 500 μm, most of the particles have a developed surface. The average size of the particles is 50–150 μm.

SEM images of representative APDDR particles consisting of microblocks are presented in Figure 1I–N. The particles contain a large number of pores and voids, i.e., the particle structure resembles a “cauliflower” in view and consists of microblocks. The size of the microblocks is 5–50 μm. Agglomerated (self-similar) structure can be noted as a specific feature of APDDR particles. The self-similarity particle is preserved for almost all sizes, including microblocks.

Microphotographs of APDDR particle segments at large magnification (1600–3000) are shown in Figure 1O–Q. As can be seen, the microblocks are connected by strands with each other and consist of smaller fragments.

The same crumb rubber, as in the first case, and SBS in the form of powder, were used as the raw materials for the co-grinding of CR and SBS copolymer by the HTSG method. Figure 2 shows images of typical particles of SBS powder. The particles of thermoplastic elastomer are comparable in size to the particles of CR and have a porous structure.

During co-grinding CR with SBS (content SBS was from 5 to 20 wt.%), a homogeneous product of black color was obtained. The optical microscope images show that the shape and morphology of the hybrid powder particles are similar to those of the APDDR (Figure 3A,C). The material is homogeneous throughout, unlike the mechanical mixture of APDDR and SBS (Figure 3B), in which white SBS particles and black APDDR particles are clearly visible. There are no visible white SBS inclusions in the hybrid powder (Figure 3C).

Analyzing the images shown in Figure 3, it is not possible to estimate the distribution of SBS in the volume of the hybrid powder; however, based on the capabilities of the high-temperature shear-induced grinding and co-grinding method, it can be assumed that the achievable level of technological mixing lies in the region of micron and a few tenths of micron [29].

### 3.2. Study of Mechanisms of Interaction between PEM Particles and Bitumen

When using powder modifiers based on APDDR or hybrid powder, their introduction into the asphalt concrete mixture is carried out at the moment of its preparation, when the mixture temperature lies in the region of 160–165 °C. To simulate the initial stage of interaction between powder elastomeric modifiers and bitumen, modified binder samples were prepared at lower temperatures (120–140 °C) at a low mixing rate (135 rpm).

Modified binders (MB) were washed with solvents on filters with pore sizes from 100–200 nm up to 2–3 μm and examined the size and morphology of the products of interaction between rubber particles and bitumen. Studying the fragments of PEM particles washed from the modified binders, prepared at different temperatures, shows that after 1 min of agitation (135 rpm), one can see a large number of particles, about 10 μm in size, which are almost absent in the starting PEM (see Figure 1B,C). The rapid formation of such microfragments is obviously connected with the fact that PEM particles, as noted above, are composed of aggregates. In addition, at least part of these aggregates is bound together by thin strands (see Figure 1D), which can quickly disintegrate in bitumen.

The presence of pores and voids in the separated microfragments leads to their further fragmentation to the submicron level, which can be clearly seen in Figure 4B,C. This fragmentation may be caused by bitumen penetrating deep into the porous particle and breaking it due to differently directed swelling pressure and precede the classical swelling associated with the penetration of solvent between rubber macromolecules, or occur concurrently with it.

In the investigation of MB samples prepared under even milder conditions (120 °C/1 min/manual stirring), we observed APDDR fragments up to 10 μm in size with signs of swelling and detachment of thin films (Figure 5). As is shown below, similar formations were also recorded in the study of the MB surface by the AFM method.

The swelling accompanied by a detachment of the swollen layer is characteristic of polymer granules swelling in low-molecular-weight liquids. Apparently, in the process of mixing hot bitumen and crumb rubber, with the diffusion of maltenes in the near-surface layer of the rubber particle, internal stresses arise due to swelling. A swollen layer exerts a compressive action on the dry core, which restricts swelling and stops the diffusion of maltene deep into the particle, and on the bitumen—cross-linked rubber boundary a gel-like structure is formed, which is characterized by a sharp edge between the already swollen layers and the still dry central core of the cross-linked rubber.

The process of rubber particles breakdown in hot bitumen may also occur with the formation of “strands” of different thickness: from 20 ÷ 100 nm to several μm (See Figure 6A,B). Note that the “strands” have a shape that is observed in filled elastomers subjected to uniaxial strain [30]. Increasing the mixing time leads to further breakdown of the particles into smaller fragments of micro- and nano levels.

Powder modifiers (APDDR and hybrid powder) have a rather strong structural heterogeneity which is due to the conditions of rubber processing by the high-temperature shear-induced grinding method. Abrupt release of shear stresses and abrupt changes in pressure and temperature when particles leave the grinding chamber of the rotary dispergator lead to the formation of internal damages: pores and voids of various sizes [5,30]. The presence of great internal damage in PEM particles causes the possibility of their further fragmentation to the submicro level, and the breakdown process may proceed practically by the “explosive” mechanism. Note that the APDDR particles broken down to submicron sizes also have an agglomerated (self-similar) structure similar to that of PEM. Images of characteristic submicron microfragments are shown in Figure 6C–E. Note that according to SEM data, we observed that the minimum size of microfragments contained in the decay particles is about 100 nm and we were able to register the decay of particles to sizes on the order of 100–200 nm. It can be assumed that these are the particles with the densest packing of rubber molecules around the filler grains—carbon black. Microfragments of this size were also found in the original APDDR particles obtained as a result of high-temperature shear-induced grinding of worn tires [5]. In this case, the decomposition into microfragments is not accompanied by black carbon emission under the temperature and time parameters of the production of the modified asphalt mixture. Production conditions of APDDR particles (high shear forces at pressures over 100 atm and temperatures of 150–180 °C) are much harsher, and carbon black is not released.

Increasing the mixing time to 40 min, even at a mixing temperature of 180 °C, does not yet lead to complete degradation and disappearance of rubber particles. Figure 6F–H shows the SEM images of rubber particles washed with solvent from such MB.

Thus, the self-similarity of agglomerated (self-similar) PEM particles and PEM breakdown fragments in bitumen up to the size of less than 100–200 nm has been shown by SEM studies. Initial interaction mechanisms of PEM with hot bitumen were revealed: rapid (up to 1 min) fragmentation of PEM particles and swelling with separation of thin films. Note that from nano- and micro-particles of the modifier gel, spatial network structures were formed.

## 4. Conclusions

Scanning electron microscopy was used to study the structure of the original rubber particles, as well as the products of their interaction with hot bitumen, in this paper. It is shown that high-temperature shear-induced grinding leads to a significant change in the size and morphology of the rubber crumb obtained by grinding on rollers. It is shown that the co-grinding of CR with butadiene styrene thermoplastic elastomer (SBS) generates optically homogeneous hybrid particles.

Self-similarity is the main structural characteristic of powder particles (rubber and hybrid) obtained by HTSG. On the first level, we can note the microblocks with a characteristic size of 5–50 μm. They are composed of smaller fragments, weakly connected to each other by “strands” of different thicknesses. The presence of a large number of pores and voids is characteristic.

The structure of particles formed by the interaction of powder elastomeric modifiers with hot bitumen in a wide temperature-time interval (120–180 °C, 1–40 min of mixing) has been studied. It was shown that multiple breakdowns of the modifier particles down to the size of 100–200 nm already take place in one minute of mixing. Micro- and submicro- fragments of broken rubber particles also have an agglomerated structure and self-similarity, similar to the original rubber particles.

Rapid fragmentation can be caused by bitumen penetrating deep into a porous particle and breaking it under the action of multidirectional swelling pressure and breaking the nano- and micro- “strands”. Fragmentation can precede the classical swelling associated with solvent penetration between rubber macromolecules or occur simultaneously with it.

Increasing the interaction time of PEM with bitumen does not lead to a marked decrease in the number of broken fragments, i.e., no complete disintegration of particles to the molecular level in the studied range of temperatures and mixing times are observed. On the contrary, with increasing time, there is a tendency for the formation of a gel spatial network.

Considering that the investigated temperature interval and the mixing time of about 1 min correspond to the technological regimes of producing various asphalt mixtures, the obtained results allow us to recommend using PEM by the “dry” process, i.e., when the modifier is introduced into the asphalt mixture while it is being produced. It is possible, of course, to prepare a modified binder under milder conditions than usual, but the “dry” process is the most economical.

Rheological studies of PEM-modified bitumen and a comparison of SEM data on the breakdown of PEM particles in hot bitumen with atomic force microscopy data will be presented in the next study.

## Figures and Tables

**Figure 1 polymers-14-03627-f001:**
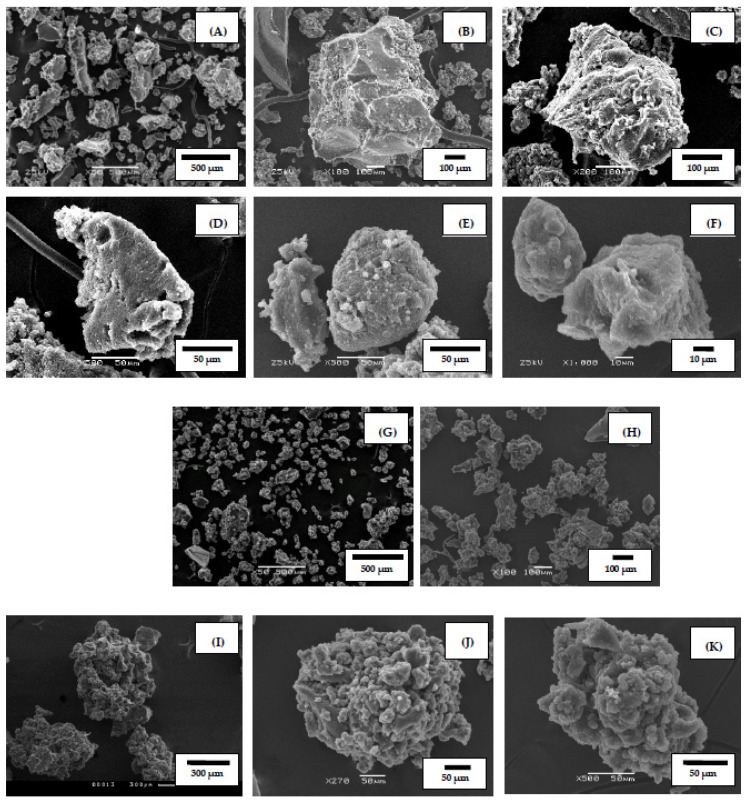
(**A**–**F**) SEM images of crumb rubber particles obtained by roll grinding; (**G**–**H**) SEM images of the general view of the APDDR particles; (**I**–**N**) SEM images of specific APDDR particles; (**O**–**Q**) SEM images of the APDDR particle segments.

**Figure 2 polymers-14-03627-f002:**
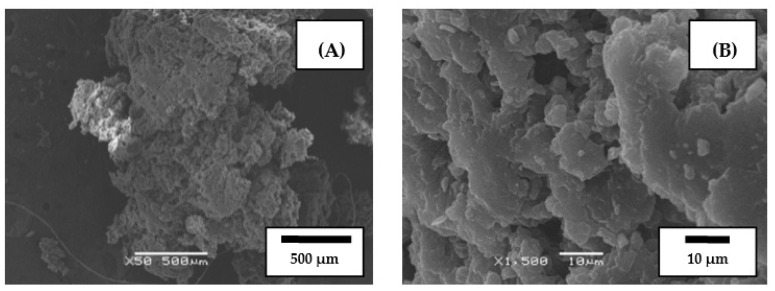
SEM images of butadiene styrene thermoplastic elastomer particle (SBS L 30-01) (**A**) and its segment (**B**).

**Figure 3 polymers-14-03627-f003:**
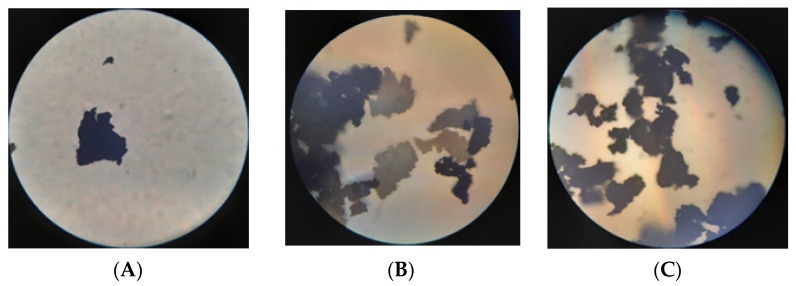
Images: (**A**) APDDR particle; (**B**) mechanical mixture of APDDR and SBS; (**C**) hybrid powder particles (optical microscope MCRT 5000 “Micros”. The magnification is 10 × 10).

**Figure 4 polymers-14-03627-f004:**
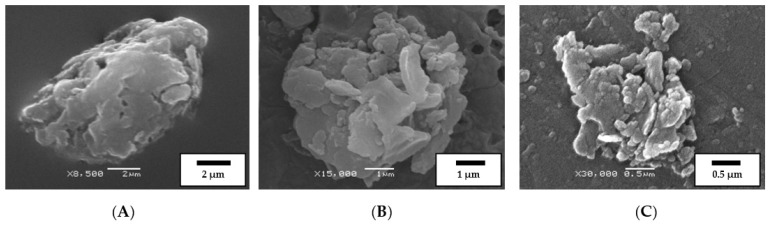
SEM images of characteristic fragments of rubber particles on the filter after washing bitumen from the modified binder with solvent. MB preparation time 1 min at 120 °C (**A**); 140 °C (**B**); 180 °C (**C**).

**Figure 5 polymers-14-03627-f005:**
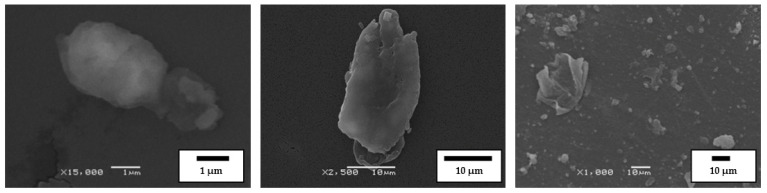
SEM images of characteristic fragments of rubber particles in the form of films on the filter after washing bitumen from MB, prepared under conditions of 120 °C/1 min/manual stirring.

**Figure 6 polymers-14-03627-f006:**
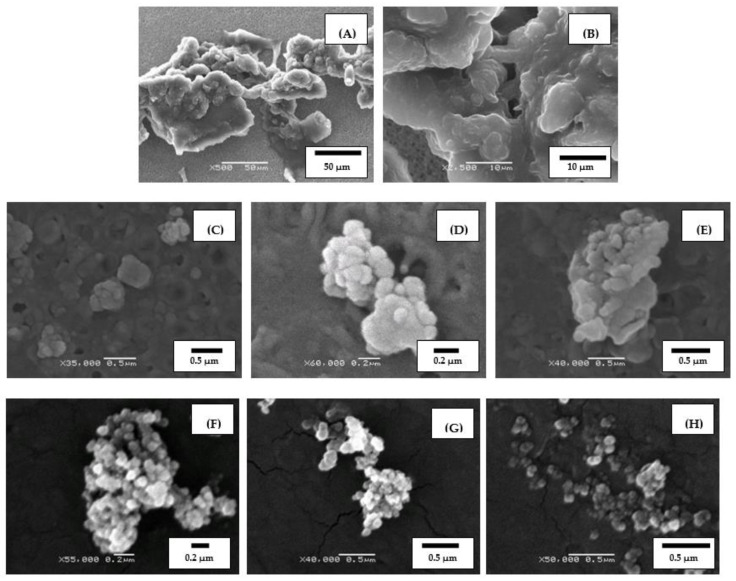
(**A**,**B**) SEM images of APDDR fragments with the formation of “strands” (after washing bitumen from MB, prepared under conditions of 140 °C/1–40 min/135 rpm); (**C**–**E**) SEM images of PEM microfragments washed with solvent from MB prepared under conditions of 140 °C/1 (3) min/135 rpm; (**F**–**H**) SEM images of PEM microfragments washed with solvent from MB prepared under mixing conditions: 180 °C/40 min.

## Data Availability

The data presented in this study are available on request from the corresponding author.

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
