# Peer review of "Ultra-Dispersed Powders Produced by High-Temperature Shear-Induced Grinding of Worn-Out Tire and Products of Their Interaction with Hot Bitumen"

_polymers, 2022, doi:10.3390/polym14173627_

Round 1
Reviewer 1 Report
The authors propose an in-depth investigation of the interaction between hot bitumen and powder elastomeric modifiers by means of electron and optical microscopy. The research topic is interesting and well-elaborated.
The manuscript is almost suitable for publication. It requires only a minor revision and some corrections listed below.
- While the study appears to be sound, many sentences are too long. Also, many “of” make reading not very fluent. Correcting these sentences would improve readability. Please check and revise. Some examples are given below:
Lines 17–20: “Structural features of particles of crumb rubber (CR) of unsorted worn-out tires obtained by grinding on rollers and ultra-disperse powder elastomeric modifiers (PEM) obtained by high-temperature shear-induced grinding (HTSG) of CR or its co-grinding with butadiene styrene thermoplastic elastomer (SBS) have been studied by methods of electron and optical microscopy.”
Lines 38–43: “The use of crumb rubber as a modifier of bitumen and asphalt concrete, taking into account the potential volumes that can be demanded by the road industry, allows to effectively combine the solution of problems of disposal of worn-out tires and increase the durability of pavement and opens the possibility of transition to a closed-loop economy, which allows to reduce the consumption of primary natural resources.”
- Introduction: the background has been well investigated. However, it would be appreciated if the goals of this work and what the authors have done to achieve them were highlighted.
Conclusions: it would be appreciated if the specific contributions to the current state of the art were emphasized.
Author Response
2022-08-25
Response to Reviewer 1
Dear Reviewer:
Thank you so much for your efforts to give our submission the best shape. From our side we tried to address all your comments (see below):
- “While the study appears to be sound, many sentences are too long. Also, many “of” make reading not very fluent. Correcting these sentences would improve readability. Please check and revise. Some examples are given below:
Lines 17–20: “Structural features of particles of crumb rubber (CR) of unsorted worn-out tires obtained by grinding on rollers and ultra-disperse powder elastomeric modifiers (PEM) obtained by high-temperature shear-induced grinding (HTSG) of CR or its co-grinding with butadiene styrene thermoplastic elastomer (SBS) have been studied by methods of electron and optical microscopy.”
Lines 38–43: “The use of crumb rubber as a modifier of bitumen and asphalt concrete, taking into account the potential volumes that can be demanded by the road industry, allows to effectively combine the solution of problems of disposal of worn-out tires and increase the durability of pavement and opens the possibility of transition to a closed-loop economy, which allows to reduce the consumption of primary natural resources.””
Lines 17-21 Structural features of crumb rubber (CR) particles obtained by grinding on rollers and ultra-disperse powder elastomeric modifiers (PEM) obtained by high-temperature shear-induced grinding (HTSG) of CR or co-grinding with butadiene styrene thermoplastic elastomer (SBS) have been studied by electron and optical microscopy methods.
38-41 The using of crumb rubber as a modifier of bitumen and asphalt mixtures in order to increase their lifespan has great potential in the road industry. Combining the problems of recycling used tires and increasing the durability of road pavements opens the transition to a closed-loop economy, which allows to reduce the consumption of primary natural resources.
72-75 rubber particles swelling in the maltene (aromatic) fraction of bitumen, i.e. the process of their volume expansion due to the absorption of light fractions from bitumen, meanwhile the penetration rate of the relevant bitumen fractions into the volume of the rubber particle is determined by the bitumen viscosity;
79-81 The bitumen chemical nature determines the equilibrium swelling, and the bitumen viscosity determines the penetration rate of the appropriate bitumen fractions into the volume of the rubber particle [24].
87-95 If the temperature of preparation of rubber-bitumen mixture is high enough and mixing time is long enough, devulcanization and depolymerization processes may lead to complete rubber disintegration, which reduces the modifier effect on bitumen properties [25, 26].
The interaction kinetics of tire rubber-based modifiers with bitumen and the size of the interaction products determine the directions for using the modifier in the road industry - by the "wet" process (a separate technological stage of preparation of modified binder), or by the "dry" process, introducing the modifier directly into the mixer at the time of preparation of asphalt mixture.
131-134 To improve the microphotography clarity and to increase the charge dripping from the material under study, a 100-300 Å layer of gold was sprayed on the surface of the samples using a "Fine coat" "JFC-1100" (Japan).
136-138 To study the products of interaction of APDDR with hot bitumen (fragments of original particles) modified binder were washed on the microfilter with solvents (Stoddard solvent, chloroform, petroleum solvent "Nefras S2-80/120").
155-157 In Figure 1 (A-F) SEM-images of crumb rubber particles obtained by roll grinding, one can see both small particles of 5÷20 μm and a significant number of large particles with sizes exceeding 500 μm.
163-165 Figure 1 (G-H) shows the general view of APDDR particles obtained by high-temperature shear-induced grinding of CR in a rotary dispergator at 140÷160 °C in the grinding zone.
224-226 In the investigation of MB samples prepared under even milder conditions (120 °C / 1 min / manual stirring), we observed APDDR fragments up to 10 μm in size with signs of swelling and detachment of thin films (Figure 5).
228-229 The swelling accompanied by detachment of the swollen layer is characteristic of polymer granules swelling in low-molecular weight liquids.
248-250 SEM images of APDDR fragments with the formation of “strands” (after washing bitumen from MB, prepared under conditions of 140 °Ð¡ / 1 - 40 min / 135 rpm);
257-260 Abrupt release of shear stresses and abrupt changes in pressure and temperature when particles leave the grinding chamber of the rotary dispergator leads to the formation of internal damages: pores and voids of various sizes [5, 30].
281-283 Initial interaction mechanisms of PEM with hot bitumen were revealed: rapid (up to 1 minute) fragmentation of PEM particles and swelling with separation of thin films.
- “Introduction: the background has been well investigated. However, it would be appreciated if the goals of this work and what the authors have done to achieve them were highlighted”
We edited Introduction and especially added the following paragraph at the end: “Development of high-temperature shear-induced grinding technology and development of new design of rotary dispergators carried out by the authors allowed to provide stable quality of powder elastomeric modifiers on industrial scale. This paper goal was to investigate the temperature-time intervals of PEM interaction with bitumen in order to work out recommendations for the using of PEM in the production of various asphalt concrete and crushed-mastic mixtures.”
- Conclusions: it would be appreciated if the specific contributions to the current state of the art were emphasized.
We edited Conclusions accordingly
Sincerely
Dr. Alex Vetcher

Reviewer 2 Report
Overall an interesting paper. The Authors have done a good job to present a study that describes the significant change in size and morphology of rubber particles processed at high-temperature shear-induced grinding.
The methods and the results presented are valid and properly supported. The approach is original and useful to research and technical purposes. The paper is also well written with clear stated objectives.
Author Response
2022-08-25
Response to Reviewer 2
Dear Reviewer:
Thank you so much for your evaluation of our submission. As about your Moderate English changes suggestion – we edited all the body of the submission, especially Introduction and Conclusions sections accordingly.
Sincerely
Dr. Alex Vetcher
Reviewer 3 Report
The manuscript entitled "Ultra-dispersed powders produced by high-temperature shear-induced grinding of worn-out tire and products of their interaction with hot bitumen", from the authors Vadim Nikol'skii, Tatiana Dudareva, Irina Krasotkina, Irina Gordeeva, Alexandre A. Vetcher, and Alexander Botin.
The use of material obtained from worn-out tires is of course very important from an ecological and economic point of view. In this manuscript, the authors offer an improved solution with the additional use of small amounts of energy. "The method of high-temperature shear-induced grinding (HTSG) allows to obtain ultra-disperse powders from CR at specific energy costs not exceeding 150 watts/kg. " What cannot be seen from this manuscript is the overall effect of adding APDDR (active powder of discretely devulcanized rubber) particles on bitumen properties. According to the authors, this will be the topic of the next study. The manuscript makes a significant contribution to the use of materials from used tires.
I consider that manuscript should be published in Polymers journal without any corrections.
Author Response
Dear Reviewer:
Thank you so much for your high evaluation of our submission.
Sincerely
Dr. Alex Vetcher